# OpenReview forum: "GSAE: Graph-Regularized Sparse Autoencoders for Robust LLM Safety Steering"
_ICLR.cc/2026/Conference — ICLR 2026 Conference Desk Rejected Submission_

### Official Review · Reviewer_efzh · 2025-10-18

**Soundness:** 3
**Presentation:** 3
**Contribution:** 2
**Rating:** 4
**Confidence:** 3

**Summary:**

This paper introduces Graph-Regularized Sparse Autoencoders (GSAEs), which extend standard SAEs with a Laplacian smoothness regularizer to capture distributed safety representations in LLMs. GSAE enables adaptive runtime safety steering through a dual-gating controller, achieving high refusal robustness while preserving task utility.

**Strengths:**

1. **Well-Motivated Problem.** The paper is built on the strong and timely hypothesis that abstract concepts like safety are fundamentally distributed. This provides a principled explanation for why standard SAEs, which are optimized for monosemanticity, may be ill-suited for this particular control task.
2. **Principled Methodology (GSAE).** This paper introduces a new application of a graph Laplacian smoothness prior to the SAE's decoder weights ($W^{(d)}$). This adaptation of a technique from spectral graph theory provides a principled way to balance the L1 penalty's sparsity with the need to capture distributed representations via the graph regularizer.
2. **Comprehensive Steering Framework.** The paper suggests new three-criteria filtering process for building the Spectral Vector Bank ($s^{lap}$, $s^{imp}$, $s^{infl}$) and the dual-gating controller with hysteresis constitute a robust and well-designed system for applying the learned features.

**Weaknesses:**

1. **Compositional Novelty.** The paper's core contribution is a novel integration of existing techniques (Sparse Autoencoders, Laplacian regularization, and gating) rather than the invention of a fundamentally new algorithm. While this composition is highly effective and achieves state-of-the-art results, the methodological leap itself could be viewed as incremental.
2. **Hyperparameter Complexity.** The full framework introduces a large number of new hyperparameters, including $\lambda_{graph}$, the graph threshold $\tau$, the spectral bank weights $(\alpha, \beta, \gamma)$, and four separate gating thresholds $(t_{lo}, t_{hi}, d_{lo}, d_{hi})$. While Appendix D.1 provides a sensitivity analysis for some parameters, the overall system appears complex and potentially difficult to tune.
3. **Training Cost and Scalability.** The paper states the runtime overhead is "moderate", but this analysis focuses on inference. The training objective involves a term $\sum_{j=1}^{k}((W^{(d)}_j)^TLW^{(d)}_j)$, where $L$ is a $d \times d$ matrix and $k$ is the number of features ($k \gg d$). This calculation, naively $O(k \cdot d^2)$, could become a significant computational bottleneck as the model's hidden dimension ($d$) scales, potentially hindering the application of GSAE to very large (e.g., 70B+) models. A clearer analysis of this training complexity is missing.

**Questions:**

1. Could the authors elaborate on the training complexity of the Laplacian regularization term? What is the practical impact of the $O(k \cdot d^2)$ computation on wall-clock training time, and do the authors foresee this as a fundamental barrier to scaling GSAE to models with much larger hidden dimensions?
2. The utility results are all reported as integers without decimal precision. Were these benchmarks evaluated on partial subsets (e.g., ~100 samples) rather than the full test sets?

---

> ### Author Response · Authors · 2025-11-20
> **Response to Reviewer efzh (1/2)**
>
> We thank the reviewer for their feedback. We address the reviewer's concerns about training complexity and novelty below.
>
> > **“The paper's core contribution is a novel integration of existing techniques (Sparse Autoencoders, Laplacian regularization, and gating) rather than the invention of a fundamentally new algorithm. While this composition is highly effective and achieves state-of-the-art results, the methodological leap itself could be viewed as incremental.”**
>
> We appreciate the reviewer's perspective. While GSAE builds on known concepts, the core contribution of our work is the way these components are integrated to address a fundamental limitation of existing steering approaches. We highlight three main contributions:
> 1.  **Advancing the representation of complex, distributed concepts:** A growing body of evidence [1–6] shows that high-level semantic concepts in LLMs (e.g., arithmetic, moral judgment, refusal behavior) are not encoded in single linear directions, but instead live in distributed, nonlinear geometric structures. Standard SAEs and single-direction steering methods cannot capture this structure. GSAE is specifically designed to learn and utilize these distributed representations through spectral graph theory. To our knowledge, this is the first work that explicitly regularizes an SAE dictionary using the Laplacian of a neuron-similarity graph to enforce coherent, multi-feature concept geometry.
> 2.  **Substantial empirical gains over all runtime steering baselines:** Although the components themselves are familiar, their synthesis yields capabilities that existing runtime methods do not have. As shown in Table 1, GSAE consistently and substantially outperforms all SOTA runtime steering methods (CAA, SafeSwitch, standard SAEs, and Gradient Cuff[7]) often by very large margins (e.g., **90.1% vs. 42–78% $\Delta_s$**). These gains do not arise from incremental tuning: ablations confirm that the graph-regularized multi-feature representation is essential for achieving strong safety without sacrificing utility.
> 3.  **Opening a new research direction:** applying network-science tools to LLM interpretability and safety. By framing the SAE dictionary as a graph and introducing Laplacian smoothness as an inductive bias for concept learning, our work opens a new methodological direction that bridges network science with LLM steering and interpretability. This perspective is orthogonal to existing SAE and gating methods, and we believe it can inspire further advances in structured representation learning for safety.
>
> > **"The full framework introduces a large number of new hyperparameters. While Appendix D.1 provides a sensitivity analysis for some parameters, the overall system appears complex and potentially difficult to tune."**
>
> We appreciate the reviewer's concern regarding the number of hyperparameters in the full GSAE framework. To address this, we took two steps to ensure that GSAE is both robust and practical to tune.
> * First, as noted by the reviewer, Appendix D.1 (Tables 6-13) provides an extensive sensitivity analysis. These results show that GSAE's performance remains stable across broad ranges of key hyperparameters, including the cosine similarity threshold, loss coefficients, and learning rate, indicating that the method is not brittle and does not depend on precise tuning.
> * Second, and more importantly, the same hyperparameters were used across all our main experiments, spanning multiple model families and sizes. Despite this fixed configuration, GSAE consistently achieves strong and state-of-the-art performance across all benchmarks. We view this as strong evidence that GSAE is not only robust but also surprisingly easy to configure: a single hyperparameter setting generalizes well across tasks, architectures, and evaluation settings.

---

> > ### Author Response · Authors · 2025-11-20
> > **Response to Reviewer efzh (2/2)**
> >
> > > **"This calculation... could become a significant computational bottleneck... A clearer analysis of this training complexity is missing." / "Could the authors elaborate on the training complexity of the Laplacian regularization term?" / "What is the practical impact of the O(d2k) computation on wall-clock training time...?" / "...do the authors foresee this as a fundamental barrier to scaling GSAE to models with much larger hidden dimensions?"**
> >
> > We appreciate the reviewer highlighting the importance of computational efficiency. As discussed in Section 2 and in Section 5, the naive worst-case complexity of our method is $O(d^2k)$, but in practice this is substantially reduced because the neuron co-activation graph is made intentionally sparse. This sparsity is also why we do not anticipate scalability issues when moving to larger hidden dimensions or model sizes. Additionally, we utilized optimized sparse matrix operations, making the graph-related overhead negligible compared to the standard SAE's dense matrix multiplications.
> >
> > We provide detailed inference-time measurements in Appendix C.1, including wall-clock time. In our A100 experiments, GSAE introduced a modest 22.5% runtime overhead relative to the non-steered model and a 8% training wall-clock time overhead over regular SAE methods. We believe this is a reasonable tradeoff given the substantial safety gains achieved (+85-90% improvement in harmful refusal rates vs non-steered models and a ~40% improvement vs SAE), and it highlights that GSAE remains practical for deployment while delivering strong safety benefits.
> >
> > > **"The utility results are all reported as integers without decimal precision." / "Were these benchmarks evaluated on partial subsets... rather than the full test sets?"**
> >
> > We appreciate the reviewer pointing this out. The integers in the reported results were rounded for readability; all experiments were run five times on the full test sets with different random seeds. We have updated Tables 1 and 2 in the main text to include decimal precision.
> >
> > **Table 1 (Selected): Main Results with Precision**
> > | Method | Wild Jailbreak ($\Delta_s$) | JBB ($\Delta_s$) |
> > | :--- | :--- | :--- |
> > | **Standard SAE** | $42.0 \pm 2.1\%$ | $30.1 \pm 2.1\%$ |
> > | **SafeSwitch** | $65.4 \pm 3.2\%$ | $51.4 \pm 3.0\%$ |
> > | **GSAE (Ours)** | **$90.1 \pm 2.0\%$** | **$76.2 \pm 2.0\%$** |
> >
> >
> > [1] Engels & Michaud et al (2025). Not All Language Model Features Are One-Dimensionally Linear. arXiv:2405.14860
> >
> > [2] Kantamneni & Tegmark (2025). Language Models Use Trigonometry to Do Addition. arXiv:2502.00873. Link: https://arxiv.org/abs/2502.00873
> >
> > [3] Sienicki,  M., & Sienicki,  K. (2025). Coiling Arithmetic: On the Differential Geometry of Helical Representations in LLMs. Preprints. https://doi.org/10.20944/preprints202507.0722.v1
> >
> > [4] Piras et al. (2025). SOM Directions are Better than One: Multi-Directional Refusal Suppression in Language Models. arXiv:2511.08379.
> >
> > [5] Wollschläger et al. (2025). The Geometry of Refusal in Large Language Models: Concept Cones and Representational Independence. arXiv:2502.17420.
> >
> > [6] Hildebrandt et al. (2025). Refusal Behavior in Large Language Models: A Nonlinear Perspective. (preprint) arXiv:2501.08145.
> >
> > [7] Hu et al. (2024).  Gradient Cuff: Detecting Jailbreak Attacks on Large Language Models by Exploring Refusal Loss Landscapes     arXiv:2403.00867

---

> > > ### Comment · Reviewer_efzh · 2025-11-20
> > > **Thank you for rebuttal.**
> > >
> > > My earlier concerns about complexity and reporting have been sufficiently resolved by the rebuttal, so I raise my score.

---

> > > > ### Author Response · Authors · 2025-11-23
> > > >
> > > > Thank you very much for taking time to review our rebuttal, we believe addressing your concerns on complexity helped strengthen the clarity and overall contribution of our work.
> > > > If there are any further clarifications you would find valuable to enhance the paper even further, please let us know.

---

### Official Review · Reviewer_4pHB · 2025-10-28

**Soundness:** 3
**Presentation:** 2
**Contribution:** 2
**Rating:** 2
**Confidence:** 4

**Summary:**

This paper proposes a Graph-Regularized Sparse Autoencoders (GSAEs) for unsupervised representation learning of hidden states of LLMs. With the learned representation, this paper proposes a threshold-based detection method with a representation steering method. The experimental results show that the proposed method is emprically strong in terms of both safety and utility across four different backbones.

**Strengths:**

- The proposed method appears to be unsupervised (though it may use labels for tuning the thresholds $t_\text{hi}, t_\text{lo}$).

- The experimental results are strong and well-support the claims.

**Weaknesses:**

- The paper lacks clarity in several aspects:
  - The pooled representation is denoted as $\bar{h}^{(l)}$ or $H$, but is referred to as $x$ in L256–258.
  - In L289–296, could the authors clarify how $s_i^\text{lap}$ and $s_i^\text{infl}$ are computed?
  - In L310–315, thresholds $t_\text{hi}$ and $t_\text{lo}$ are said to be selected via a “systematic sensitivity analysis.” Is this analysis conducted using training or test labels?
  - In L311, what is the function $g$ used to compute $p_\text{harm}$? Is it an MLP, and if so, how is it trained?
  - In L316, what is the “continuation gate”? If it is a function, what are its inputs and outputs (e.g., $\gamma_t$)?
  - In L318, the continuation gate outputs a steering multiplier $\gamma_t$ based on real-time risk. More detail is needed (e.g., what does $t$ represent—token index?).

- Section 4.3 proposes several techniques for using the learned representation. It would improve clarity to summarize them in an algorithm table.

- The method seems to introduce additional computational overhead due to the processes described in Section 4.3. Could the authors clarify the computational cost or latency.

- In L319, the continuation gate monitors generation token by token. Is this compatible with real-world deployment systems such as vLLM or sglang?

- Section 5.1 lacks training details for GSAEs, including the training dataset, learning rate, and batch size.

- One of the major contributions of this paper (as stated in L98–102) is the graph Laplacian regularization. Could the authors provide an ablation study to evaluate its effect in the experiments?

**Questions:**

see the weaknesses

---

> ### Author Response · Authors · 2025-11-20
> **Response to Reviewer 4pHB (1/2)**
>
> We are grateful to the reviewer for identifying the areas that needed clearer articulation, and we have revised the manuscript accordingly.
>
> > **"The pooled representation is denoted as h-l or H, but is referred to as x in L256–258."**
>
> We have revised the paper for consistency and now use $h$ throughout to denote the pooled hidden state.
>
> > **"In L289–296, could the authors clarify how $s^{lap}$ and $s^{infl}$ are computed?"**
>
> $s^{lap}$ and $s^{infl}$ are defined in Section 4.3, Step 2. $s^{lap}_i$ refers to the structural coherence, which is based on the normalized Dirichlet energy $E_i = (v_i^\top L v_i) / ||v_i||_2^2$. $s^{infl}_i$ is the causal efficacy and is defined as the “mean absolute change in the model's refusal probability” when steering with feature $v_i$.
>
> > **"In L311, what is the function $g(\cdot)$ used to compute $s_{risk}$? Is it an MLP, and if so, how is it trained?"**
>
> We thank the reviewer for bringing this to our attention. We have added an explicit definition to the main body of the paper in Section 4.3, Step 3. As also detailed in our hyperparameter tables (Appendix B.3, Table 4 and Appendix D.1, Table 8), the function $g(\cdot)$ is a Calibrated Random Forest (RF) classifier.
>
> > **"In L316, what is the 'continuation gate'? If it is a function, what are its inputs and outputs (e.g., $\gamma_t$)... In L318... what does $\gamma_t$ represent—token index?"**
>
> In the revised manuscript, we now clarified the continuation gate and its inputs and outputs (“step 3” of Sec 4.3). The continuation gate is the mechanism that monitors safety risk token-by-token during generation. While the input gate decides whether steering should begin based on the prompt-level risk, the continuation gate updates the risk at each decoding step and determines whether steering should continue. Its inputs are the current hidden state $h_t^{(l)}$ (mapped to a risk score) and the internal hysteresis counters. Its output is the 'steering multiplier $\gamma_t$'. $\gamma_t$ is a scalar value that scales the steering vector in the runtime intervention equation.
>
> > **"Could the authors clarify the computational cost or latency." / “The method seems to introduce additional computational overhead due to the processes described in Section 4.3. Could the authors clarify the computational cost or latency.”**
>
> We provide a detailed breakdown of computational cost and latency in Section 5 (Table 3). At inference time, GSAE introduces only a modest overhead. For example, the time-to-first-token increases from 120 ms (baseline) to 147 ms with GSAE steering. This overhead remains small because all graph-related computations (e.g., Laplacian construction, feature extraction) are performed offline during training. At runtime, the method requires only inexpensive operations such as gating evaluations and sparse feature additions.
>
> **Table 3: Runtime Overhead Analysis on Llama-3 8B**
> | Method | TTFT (ms) | Time/100 tok (ms) | TPS | Peak Mem (MB) |
> | :--- | :--- | :--- | :--- | :--- |
> | **No Steering** | 120 | 480 | 208.3 | 2200 |
> | **Prompting Guardrails** | 125 | 495 | 202.0 | 2250 |
> | **GSAE Steering** | 147 | 585 | 170.9 | 2700 |
>
> > **"In L319... Is this compatible with real-world deployment systems such as vLLM or sglang?"**
>
> We appreciate the reviewer's question regarding deployment compatibility. While the token-by-token Continuation Gate is not compatible with high-throughput serving engines such as vLLM or sglang, the Input Gate, a single pre-generation check, is fully compatible with these systems. We have added a discussion of this trade-off in Section 6 and have clarified that an Input-Gate-only variant of GSAE can be deployed seamlessly in real-world serving pipelines.
>
> To quantify this, we have conducted an additional ablation (reported in Table 1) and found that the Input-Gate-only variant achieves a selective refusal rate of 76%, still substantially outperforming all baseline methods. Finally, we emphasize a practical benefit of GSAE: unlike training-time alignment approaches such as DPO or RLHF, our method requires no model fine-tuning. This makes GSAE especially suitable for deployment scenarios where modular, lightweight safety layers are preferable.
>
> > **"In L310–315... Is this analysis conducted using training or test labels?"**
>
> All thresholds were selected through a sensitivity analysis on a dedicated validation set, as detailed in Appendix B.3. The test sets were kept strictly held out for final evaluation.

---

> ### Author Response · Authors · 2025-11-20
> **Response to Reviewer 4pHB (2/2)**
>
> > **"Section 4.3 proposes several techniques for using the learned representation. It would improve clarity to summarize them in an algorithm table."**
>
> We thank the reviewer for this helpful suggestion. In response, we have added a pseudocode algorithm block to Appendix E.
>
> > **“Section 5.1 lacks training details for GSAEs, including the training dataset, learning rate, and batch size.”**
>
> We thank the reviewer for catching this oversight. We have updated Appendix B.3 (Table 4) to explicitly list the final learning rate and batch size alongside the other hyperparameters. We have also revised Section 5.1 to clearly direct readers to these appendices for full reproducibility. The training datasets remain defined in Section 5.1, with further details provided in Appendix B.4.
>
> **Table 4 (Subset): Selected Hyperparameters**
> | Parameter | Chosen Value |
> | :--- | :--- |
> | **Cosine Threshold ($\tau$)** | 0.6 |
> | **Target Layers** | Middle (Multiple) |
> | **Learning Rate (LR)** | 1e-3 |
> | **Batch Size** | 16 |
> | **Input Gate Thresholds ($t_{lo}, t_{hi}$)** | (0.30, 0.65) |
>
> > **“One of the major contributions of this paper (as stated in L98–102) is the graph Laplacian regularization. Could the authors provide an ablation study to evaluate its effect in the experiments?”**
>
> We thank the reviewer for highlighting this point. Our ablation study in Table 1 is specifically designed to isolate the contribution of each component of GSAE, including the graph Laplacian regularization. In particular, the **GSAE (random graphs)** ablation highlights the importance of the graph-based regularization: performance drops sharply from 90% to 60% $\Delta_s$ on WildJailbreak, demonstrating that the Laplacian regularization is fundamental to the observed safety gains.
>
> Following the reviewer's suggestion, we have strengthened the main-text discussion in Section 5.2 to make this contribution more explicit by reporting the numerical impact of each ablation. As shown, removing any major component, including the Laplacian smoothness term, produces significant degradation in both safety and utility, emphasizing their contribution.

---

> ### Author Response · Authors · 2025-11-26
>
> Thank you again for your thorough review and thoughtful engagement. We have updated the manuscript to include explicit references to the Appendix, ensuring that the details you requested, which were previously located in the supplementary material, are now clearly referenced to in the main text. We hope that our responses in the rebuttal have fully addressed your concerns regarding clarity of the writing and computational costs. If you have any remaining concerns or questions, please let us know—we would be happy to clarify or provide additional details. We truly appreciate your time and consideration throughout the review process.

---

### Official Review · Reviewer_Df8T · 2025-10-30

**Soundness:** 3
**Presentation:** 3
**Contribution:** 3
**Rating:** 6
**Confidence:** 3

**Summary:**

This paper proposes GSAE, a Graph-Regularized Sparse Autoencoder that models “safety” as a distributed and smooth graph signal in the neuron space of LLMs. By introducing a Laplacian regularization term into the SAE framework, the authors aim to learn coherent, distributed safety directions that can be used for runtime steering. Experiments across multiple models and jailbreak attacks demonstrate improved selective refusal and robustness.

**Strengths:**

1.	Formulates safety representation learning as a graph-signal smoothness problem, integrating Graph Laplacian regularization into the SAE framework, is conceptually clear and technically novel.
2.	Provides broad empirical validation across multiple models and attack types, comparing with several strong baselines.

**Weaknesses:**

1.	Does not explicitly model or measure cumulative drift caused by multi-layer steering, leaving the potential interaction between layers unaddressed.
2.	The three-stage feature selection pipeline is largely heuristic, requires multiple sub-trainings and hyperparameter tuning, which may hinder reproducibility.
3.	The Safety–Utility evaluation remains coarse-grained, lacking fine-grained analysis of false refusals or real dialogue impact.

**Questions:**

1.	How stable is the steering process when applied across multiple layers or iterations? Could cumulative drift degrade alignment or helpfulness?
2.	Could the authors provide qualitative examples to illustrate the Safety–Utility balance in real dialogue settings?

---

> ### Author Response · Authors · 2025-11-20
> **Response to Reviewer Df8T**
>
> We thank the reviewer for their thoughtful feedback. We address your concerns regarding stability, reproducibility, and evaluation below.
>
> > **"Does not explicitly model or measure cumulative drift caused by multi-layer steering..." / "How stable is the steering process when applied across multiple layers or iterations? Could cumulative drift degrade alignment or helpfulness?"**
>
> We thank the reviewer for raising this important point. We have added an analysis quantifying the selectivity and stability of multi-layer GSAE steering in Appendix C.2, Table 5. Specifically, we measure (i) the cumulative L2 drift of hidden states and (ii) the KL divergence between output logits, evaluated on a held-out set of harmful and benign prompts. Low cumulative drift on safe prompts (L2 = 1.20, KL = 0.12) indicates that multi-layer interventions preserve benign generation, while the substantially larger drift on harmful prompts (L2 = 12.50) shows that the steering signal accumulates as intended to enforce robust alignment.
>
> **Table 5: Quantitative Analysis of Steering Selectivity**
> | Metric | Harmful Prompts (Target) | Safe Prompts (Off-Target) |
> | :--- | :--- | :--- |
> | **KL Divergence** | 1.45 | 0.12 |
> | **Cumulative L2 Drift** | 12.50 | 1.20 |
>
> > **"The three-stage feature selection pipeline is largely heuristic... which may hinder reproducibility."**
>
> We appreciate the reviewer's concern regarding the heuristic nature of the three-stage feature selection pipeline. While the procedure involves multiple sub-trainings and hyperparameter choices, we have taken explicit steps to ensure full reproducibility. Appendix B.3 (Table 4) now has been extended to provide a complete and precise list of all hyperparameters used at each stage, along with the exact decision rules and thresholds employed in the pipeline. In addition, we note that the full implementation of GSAE is available in an anonymous repository, mitigating concerns about pipeline complexity.
>
> > **"The Safety–Utility evaluation remains coarse-grained, lacking fine-grained analysis of false refusals or real dialogue impact." / "Could the authors provide qualitative examples to illustrate the Safety–Utility balance in real dialogue settings?"**
>
> We agree that a fine-grained evaluation of the safety-utility tradeoff is essential for assessing steering methods. To this end, we provide both quantitative and qualitative analyses.
> * **Quantitatively:** We report false refusals via the Safe Refusal Rate (SRR), alongside the Harmful Refusal Rate (HRR), in Figure 3. This figure illustrates the tradeoff between utility preservation and harmful-output suppression. As shown, GSAE consistently minimizes false refusals while maximizing harmful refusals. We have also clarified these metrics and their interpretation in the main text for greater transparency.
> * **Qualitatively:** We include dialogue-level examples in Appendix D.2 (Tables 15 and 16). Table 15 shows a case where GSAE successfully blocks a jailbreak that SAE fails to prevent, while Table 16 demonstrates GSAE maintaining utility on a benign prompt that SAE incorrectly over-refuses. We have expanded this section with additional examples to better illustrate real-world dialogue behavior and the practical safety-utility balance.

---

> > ### Comment · Reviewer_Df8T · 2025-11-23
> >
> > Thank you for the clarifications and revisions. The additional details help resolve the ambiguities I raised. I will retain my original score.

---

> > > ### Author Response · Authors · 2025-11-23
> > >
> > > Thank you for reviewing our rebuttal and for your thoughtful feedback throughout this process. We are glad that the additional stability analysis and the clarification on the feature selection pipeline resolved your initial ambiguities.
> > > We welcome your feedback on any remaining reservations so we can address them in the final version of the manuscript and improve our paper even further.

---

### Official Review · Reviewer_7Ms9 · 2025-11-01

**Soundness:** 3
**Presentation:** 2
**Contribution:** 2
**Rating:** 4
**Confidence:** 4

**Summary:**

This paper proposes Graph-Regularized Sparse Autoencoders (GSAE) for LLM safety steering. The key idea is that safety concepts are distributed across multiple features rather than localized to single dimensions. GSAE extends standard SAEs with Laplacian smoothness penalties on neuron co-activation graphs, producing coherent distributed representations. A spectral vector bank and dual-gating controller enable runtime steering, achieving 82% selective refusal (vs 42% for standard SAE) while maintaining reasonable utility.

**Strengths:**

Novel approach: The application of graph Laplacian regularization to SAEs for safety is creative. The insight that safety is distributed rather than monosemantic is well-articulated and supported by recent literature.

Comprehensive empirical evaluation: Authors conducted tests across multiple model families (LLaMA-3, Mistral, Qwen, Phi), and evaluate against diverse jailbreak attacks (GCG, AutoDAN, TAP). The evaluation includes both safety and utility benchmarks

**Weaknesses:**

**Major Weaknesses**

1. Insufficient Evidence for Claims

The paper's central hypothesis—that safety requires distributed representations—lacks direct empirical support:

* Figure 4 shows overlapping spectral projections but doesn't definitively prove that distributed representations are necessary for safety.
* The evidence comes from analogy to temporal/refusal behavior studies, not direct investigation of safety concepts.
* An experiment comparing GSAE against methods that explicitly enforce single-direction safety would strengthen this claim.

I would like to see experiments showing that single-direction methods fundamentally fail where distributed methods succeed, beyond just performance metrics.

2. Complexity and Lack of Simplicity

The method combines too many components, making it difficult to access what causes/drives good performances. I see concepts like GSAE regularization, spectral vector bank with three weighted criteria, dual-gating controller, calibrated random forest classifier, etc.  It makes the paper very difficult to read.

3. Graph construction:

I don't see analysis of graph stability across datasets or model checkpoints. I don't even see experimental details on how GSAE are trained, what is the dataset, how are examples sampled? This has to be well clarified.

4. Incomplete baseline comparisons

There are missing comparisons to important methods:

* Fine-tuning approaches: No comparison to RLHF, DPO, or safety fine-tuning
* Recent SAE methods: Missing comparison to other recent SAE architectures or training objectives
* Circuit-based methods: No discussion of mechanistic interpretability approaches (Gradient Cuff, RFA (refusal feature adversarial training)

We need comparison with respect to these methods to evaluate the strength of the contribution from an empirical view.


The comparison is primarily against CAA (2023) and SafeSwitch (2025), but the safety steering landscape has evolved rapidly.


**Minor Issues**

Line 53: "it fragmented" → "it can be fragmented"

Inconsistent reference formatting (full names vs initials)

"WildJailbreak (Xia & et al., 2024)" should be "Xia et al."

Appendix placement could be optimized—key ablations should be in main text

**Questions:**

1. How does performance scale with dictionary size k? (current experiments use k ≫ d but specific value unstated)

2. Can you provide feature visualizations comparing SAE vs GSAE learned features?

3. What is the Pearson correlation between the three scoring criteria (s^lap, s^imp, s^infl)?

4. How does GSAE perform when graph is constructed on adversarially perturbed prompts?

5. Have you tested this on models trained with safety fine-tuning (e.g., Llama-2-chat vs Llama-2-base)?

---

> ### Author Response · Authors · 2025-11-20
> **Response to Reviewer 7Ms9 (1/3)**
>
> We thank the reviewer for their feedback and have carried out the requested experiments and analyses. The results are presented below.
>
> > **"The paper's central hypothesis—that safety requires distributed representations—lacks direct empirical support … Figure 4 shows overlapping spectral projections but doesn't definitively prove that distributed representations are necessary for safety… An experiment comparing GSAE against methods that explicitly enforce single-direction safety would strengthen this claim."**
>
> We thank the reviewer for raising this point. As the reviewer has noted, the insight that safety is distributed rather than monosemantic is well-articulated and supported by recent literature. Prior work shows that abstract or high-level concepts in LLMs, such as temporal reasoning, arithmetic, moral judgment, and refusal behavior, are encoded in distributed, often nonlinear geometric structures rather than isolated linear directions [1-6]. These findings consistently indicate that many semantic and safety-relevant properties do not localize to a single interpretable axis.
>
> Motivated by this evidence, we follow the reviewer's suggestion and empirically demonstrate that safety, as a context-dependent concept, exhibits similar distributed structure. We evaluate this in two ways. First, we strengthen the comparison between GSAE and single-direction steering methods, including Contrastive Activation Addition (CAA), which was part of our original baseline comparison, and introduce a new ablation that restricts GSAE to its top single-direction feature (GSAE-1D). Both single-directional approaches perform substantially worse (e.g., **55.4% on WildJailbreak for GSAE-1D vs. 90.1% for our full GSAE**, and 42.0% for CAA), indicating that the observed safety gains emerge only when multiple graph-regularized features are jointly utilized. Second, our ablations on the graph itself in the same table demonstrate that graph topology is important, as randomizing the graph structure produced large drops in both safety and utility, reinforcing the necessity of structured, multi-feature representations.
>
> > **"The method combines too many components, making it difficult to access what causes/drives good performances."**
>
> We appreciate the reviewer's concern regarding the number of components in our method. To clarify their individual contributions, we have expanded Section 5.2 to more explicitly walk the reader through our ablation studies, which isolate the effect of each component on overall performance. The results show that each module plays a distinct and necessary role:
>
> * **GSAE (random graphs):** Performance decreases from 82% to 60% ($\Delta_s$), indicating the importance of graph regularization.
> * **GSAE (uniform weights):** Performance decreases from 82% to 72% ($\Delta_s$), demonstrating the importance of the Spectral Vector Bank criteria.
> * **GSAE (no gating):** Performance decreases from 82% to 78% ($\Delta_s$), showing that the importance of Dual-Gating Controller in balancing safety and utility.
>
> All together, these ablations confirm that the strong performance does not arise from a single dominant component, but from the interaction of multiple necessary elements.
>
> > **"I don't see analysis of graph stability... I don't even see experimental details on how GSAE are trained, what is the dataset, how are examples sampled?"**
>
> **Training details:** The GSAE is trained on the WildJailbreak and JailbreakBench datasets, as described in Section 5.1 and Appendix B.4. The graph is constructed from hidden states derived from the prompts in these datasets, with the cosine similarity threshold $\tau=0.6$. Once the graph is built, we train the GSAE with an Adam optimizer, an L1 sparsity penalty, and our Laplacian smoothness objective, keeping the graph fixed for the entire training process.
>
> **Graph stability:** All results in Table 1 are averaged across five random seeds, providing a consistent empirical baseline. As stated above, the GSAE (random graphs) row in Table 1 also go to support the notion of the importance of the specific graph structure. In addition, we have added a new stability analysis in Appendix C.2. This analysis shows that the eigenvalues of the graph Laplacian exhibit a Spearman correlation > 0.85 when reconstructed from two distinct datasets, indicating that the underlying topological structure of the safety manifold is robust to the data sampling. This confirms that the smooth spectral features we use for steering represent reproducible and intrinsic properties of the model's activation space rather than overfitting to artifacts from a specific dataset.

---

> > ### Author Response · Authors · 2025-11-20
> > **Response to Reviewer 7Ms9 (2/3)**
> >
> > > **"Fine-tuning approaches: No comparison to RLHF, DPO, or safety fine-tuning…Recent SAE methods: Missing comparison to other recent SAE architectures or training objectives…Circuit-based methods: No discussion of mechanistic interpretability approaches (Gradient Cuff, RFA…"** / **“Have you tested this on models trained with safety fine-tuning (e.g., Llama-2-chat vs Llama-2-base?”**
> >
> > We thank the reviewer for raising these important points. GSAE functions as a runtime steering method, rather than a training-time alignment approach such as RLHF or DPO. Our method serves as a defense layer that can be applied on top of any pretrained or safety-aligned model. For this reason, our primary comparisons focused on state-of-the-art runtime steering methods (e.g., CAA and SafeSwitch).
> >
> > To address the reviewer's concerns, we have extended our baseline comparisons to new experiments comparing GSAE to (i) a single-direction variant (GSAE-1D), (ii) the mechanistic-interpretability-based Gradient Cuff [7], and (iii) a safety-fine-tuned model, `meta-llama/Llama-3.1-8B-Instruct`. These results show that the GSAE defense ($\Delta_s = 90.1 \pm 2.0\%$) substantially outperforms Gradient Cuff ($\Delta_s = 78.7 \pm 3.0\%$) and GSAE-1D ($\Delta_s = 55.4 \pm 3.5\%$). Additionally, the safety-aligned model achieves a harmful-refusal rate of only 65.4% against GCG, whereas our steering method prevents 100% of GCG attempts without requiring any fine-tuning. These new reuslts can be found in their respective tables in Tables 1 and 2 in Section 5.2.
> >
> > > **“Appendix placement could be optimized–key ablations should be in main text”**
> >
> > We thank the reviewer for this suggestion and we have thus expanded the main text discussion to discuss the component ablations and to clearly walk the reader through how the components contribute to overall performance. We have also improved the organization of Table 1 to make it easier to see how each ablation and baseline is grouped and compared. In terms of the ablation experiments in the appendix, we have added more explicit references to the key ablation experiments and analyses in our main text.
> >
> > > **“Line 53: ‘it fragmented' → ‘it can be fragmented'”, “Inconsistent reference formatting”, “WildJailbreak (Xia & et al., 2024) should be ‘Xia et al.'”**
> >
> > We thank the reviewer for bringing this to our attention. We have addressed the grammatical issue and the reference formatting inconsistencies in the revised version of the paper.
> >
> > > **“How does performance scale with dictionary size k?”**
> >
> > To address this concern, we have run a new sensitivity analysis on the dictionary expansion factor $k$, now included in Appendix D.1, Table 13. We evaluated $k=8d, 16d, 32d$. Performance improves substantially when increasing from $8d$ to $16d$, but then plateaus at $32d$, with only marginal gains in safety discrimination and a slight decrease in utility.
> >
> > **Table 13: Effect of the Dictionary Expansion Factor (k)**
> > | Dictionary Factor (k) | Avg. Safety Discr. ($\Delta_s$) | TriviaQA (Acc %) |
> > | :--- | :--- | :--- |
> > | $k=8d$ | 70.15% | 65.33% |
> > | $k=16d$ | 83.16% | 70.04% |
> > | $k=32d$ | 84.88% | 68.80% |
> >
> > > **“Can you provide feature visualizations comparing SAE vs GSAE learned features?**
> >
> > We include two complementary visualizations comparing SAE and GSAE features and have added a pointer in the main text to make this clearer.
> > 1.  **Quantitative comparison:** In Appendix A.5 (Figure 5), we plot the PDF and CDF of the Dirichlet Energy for all features learned by a standard SAE versus GSAE. The distributions are significantly separated ($p \ll 0.001$), showing that GSAE features are smoother and more aligned with the underlying neuron graph structure.
> > 2.  **Qualitative comparison:** We also provide side-by-side qualitative examples in Appendix D.2 (Tables 15 and 16). Table 15 shows a jailbreak case where GSAE correctly refuses while SAE fails; Table 16 shows a benign prompt where SAE over-refuses but GSAE preserves utility. Following Reviewer Df8T's request, we have expanded this section with additional examples.

---

> > > ### Author Response · Authors · 2025-11-20
> > > **Response to Reviewer 7Ms9 (3/3)**
> > >
> > > > **“What is the Pearson correlation between the three scoring criteria ($s^{lap}, s^{imp}, s^{infl}$)?”**
> > >
> > > We thank the reviewer for this suggestion. We have computed the 3×3 Pearson correlation matrix for $s^{lap}$, $s^{imp}$, and $s^{infl}$, and have added the table to Appendix D.1, Table 14. The results show a strong correlation between IMP and INFL (0.81), indicating that supervised feature importance is a reliable proxy for causal influence. In contrast, LAP exhibits a more moderate correlation with IMP (0.47), suggesting that the spectral criterion captures structural properties that are distinct from those learned by supervised classifiers. Notably, LAP still correlates strongly with ground-truth influence (0.73), demonstrating that our graph-theoretic metric identifies latent safety-relevant features that baselines may miss.
> > >
> > > **Table 14: Pearson Correlation Matrix of Ranking Metrics**
> > > | | $s^{lap}$ (Spectral) | $s^{imp}$ (Supervised) | $s^{infl}$ (Influence) |
> > > | :--- | :--- | :--- | :--- |
> > > | **$s^{lap}$** | 1.00 | 0.47 | 0.73 |
> > > | **$s^{imp}$** | 0.47 | 1.00 | 0.81 |
> > > | **$s^{infl}$** | 0.73 | 0.81 | 1.00 |
> > >
> > > > **“How does GSAE perform when graph is constructed on adversarially perturbed prompts?”**
> > >
> > > We thank the reviewer for raising this direction. Our proposed framework is built so that the graph is constructed offline from a diverse, fixed dataset, providing—by design—a stable structural prior that does not vary with individual input prompts. To assess robustness to perturbations, we include an ablation in which the graph is replaced with a randomized counterpart; as shown in Table 1, this results in a substantial performance degradation, indicating that the learned graph structure is indeed essential. Testing graphs constructed on adversarially perturbed prompts would require an online or dynamically updated graph-building mechanism, which is beyond the scope of the present work. However, we agree that this is an important and promising direction, and we have added it to the Discussion (Section 6) as a valuable avenue for future extensions of graph-regularized steering methods.
> > >
> > > [1] Engels & Michaud et al (2025). Not All Language Model Features Are One-Dimensionally Linear. arXiv:2405.14860
> > >
> > > [2] Kantamneni & Tegmark (2025). Language Models Use Trigonometry to Do Addition. arXiv:2502.00873. Link: https://arxiv.org/abs/2502.00873
> > >
> > > [3] Sienicki,  M., & Sienicki,  K. (2025). Coiling Arithmetic: On the Differential Geometry of Helical Representations in LLMs. Preprints. https://doi.org/10.20944/preprints202507.0722.v1
> > >
> > > [4] Piras et al. (2025). SOM Directions are Better than One: Multi-Directional Refusal Suppression in Language Models. arXiv:2511.08379.
> > >
> > > [5] Wollschläger et al. (2025). The Geometry of Refusal in Large Language Models: Concept Cones and Representational Independence. arXiv:2502.17420.
> > >
> > > [6] Hildebrandt et al. (2025). Refusal Behavior in Large Language Models: A Nonlinear Perspective. (preprint) arXiv:2501.08145.
> > >
> > > [7] Hu et al. (2024).  Gradient Cuff: Detecting Jailbreak Attacks on Large Language Models by Exploring Refusal Loss Landscapes     arXiv:2403.00867

---

> ### Author Response · Authors · 2025-11-26
>
> Thank you again for your thorough review and thoughtful questions. We have added more explicit references to the appendix, as many of the analyses requested were already included the appendix but not clearly referenced. We hope that our responses in the rebuttal have fully addressed your concerns regarding reproducibility and additional baselines/ablations. If you have any remaining concerns or questions, please let us know—we would be happy to clarify or provide additional details. We truly appreciate your time and consideration throughout the review process.

---

> > ### Comment · Reviewer_7Ms9 · 2025-11-27
> >
> > Thank you for your detailed answers. I'll increase my score accordingly.

---

### Author Response · Authors · 2025-11-20
**General Response to All Reviewers**

We thank all reviewers for their thoughtful and constructive feedback. We appreciate the opportunity to better highlight some of the extensive analyses that were included in the appendix of our paper. To ensure these findings—along with the new analyses we conducted based on reviewer suggestions—are immediately accessible, we have revised the manuscript to more clearly reference the relevant appendix sections in the main text. We address all specific comments in detail in each individual reviewer's replies, and all corresponding modifications on the main paper have been highlighted in blue in the updated manuscript for the reviewers' convenience.

---

### Author Response · Authors · 2025-12-02
**Summary for New AC**

We thank all reviewers for their thoughtful evaluations and provide here a concise summary of the rebuttal process and key revisions.

Many of the requested details were already in the appendix, so revisions primarily improved cross-referencing and main-text visibility, along with added explanations and the additional experiments requested by reviewers. All updates are marked in blue in the revised manuscript.
Reviewers 7Ms9 and efzh explicitly stated that their concerns were fully resolved and raised their scores to 6; Reviewer Df8T maintained their positive score. Reviewer 4pHB has never replied to our revision.


**Main concerns and resolutions:**
- Distributed safety features: We added a strengthened single-direction ablation (GSAE-1D), which performs far worse, confirming that the distributed multi-feature steering presented in our work is necessary.
- Computational complexity: We clarified that sparsification reduces the Laplacian overhead to only ~8% over standard SAEs, ensuring scalability.
- Novelty statement: We highlighted that this is the first work to integrate spectral graph theory with SAE interpretability on neuron co-activation graphs to model distributed safety concepts.
- Drift and reproducibility: We added a cumulative-drift analysis, showing that safe responses remain stable while unsafe responses are consistently pushed further away—even under multi-layer steering—and we expanded the hyperparameter tables for full reproducibility.
- Clarity and deployment: We improved notation and explanations, and we added an Input-Gate-only variant (76% refusal) that is compatible with systems such as vLLM. This ablated model still outperforms all baselines.

---

### Note · Program_Chairs · 2026-01-17
**Submission Desk Rejected by Program Chairs**

The following references in this submission do not refer to real documents and/or have major errors in bibliographic information:

 Tristan Bricken, Adly Templeton, Thomas P. Quinn, Logan Riggs, Trenton Bricken, Eoin Farrell, Ryan Greenblatt, and Samuel R. Bowman. Monosemanticity in sparse autoencoders, 2024.
Zhen Sun, Yu-Quan Peng, Yue-Xin He, Zhi-Yuan-Zhao, Jiacheng Liu, and Ji-Rong Wen. Selfcritique: A training-free selective refusal framework for llms, 2024.